# Sexual Behaviour and Fantasies in a Group of Young Italian Cohort

**DOI:** 10.3390/jcm10194327

**Published:** 2021-09-23

**Authors:** Marina Di Mauro, Giorgio Ivan Russo, Gaia Polloni, Camilla Tonioni, Daniel Giunti, Gianmartin Cito, Bruno Giammusso, Girolamo Morelli, Lorenzo Masieri, Andrea Cocci

**Affiliations:** 1Urology Section, Department of Surgery, University of Catania, 95100 Catania, Italy; marinadimauro@live.it; 2Centre of Pshycology, Via Cadorna, 22100 Como, Italy; dott.gaiapolloni@gmail.com; 3Centro Integrato di Sessuologia Il Ponte, 50100 Florence, Italy; camilla.tonioni@stud.unifi.it (C.T.); danielgiunti@gmail.com (D.G.); 4Department of Urology, University of Florence, 50100 Florence, Italy; gianmartin.cito@gmail.com (G.C.); lorenzo.masieri@meyer.it (L.M.); cocci.andrea@gmail.com (A.C.); 5Morgagni Hospital, 95100 Catania, Italy; bgiammusso@hotmail.it; 6Urology Section, University of Pisa, 56121 Pisa, Italy; girolamomorelli@gmail.com

**Keywords:** sex aid, sexual behavior, pornography, alcohol, erectile dysfunction

## Abstract

Over the years, sexual behaviour has changed due to the growing interest in everything related to the sexual sphere. The purpose of the study was to collect information on the sexual habits and behaviours of Italian people of all ages, sexes and sexual orientations and to describe the patterns of sexual behaviour, with the aim of gaining a representative picture of sexuality in Italy, before the COVID-19 pandemic. Participants completed a survey with 99 questions about their sexual habits. In our group first sexual experiences occurred on average around the age of 15, whilst the median age of the first sexual intercourse was 17. The fantasies that most stimulated and excited our group (Likert scale ≥ 3) was having sex in public (63.9%), having sex with more than one person at the same time (59.4%), blindfolded sex (64.9%), being tied up (56.3%) and observing a naked person (48.6%). As for pornography, we have shown that 80% of our group watched porn at home, alone or from their smartphones. Our results have several practical implications for the areas of sex education and sexual health. It is necessary to safeguard the health of young people and support them increasing their sexual well-being.

## 1. Introduction

Sexual behavior has consistently changed over the years. In fact, there is a growing interest in specific topics that have been considered a stimga in the past, and also physicians should constantly pay attention to patients’ preferences [1,2].

Scientific evidences have underlined that the majority of information on sexuality collected through the internet by young people regard explicit messages including and facilitating sexual practices like autoerotism and masturbation [3]. 

Exploring patterns of current sexual behaviours is important for several reasons, but mainly because the description of behavioural trends can provide an important empirical context for examining the associations between patterns of emerging sexual behaviour and aspects of sexual health and well-being among young people [4]. A recent survey by Herbenick et al. (2020) observed that more frequent past-year pornography use and a greater lifetime range of pornography accesses were significantly associated with engaging in both dominant and target sexual behaviors among all participants [5]. Furthermore, sex aids are also considered tools to help individuals achieving sexual pleasure and can also be particularly helpful for sexual dysfunction [6]. Indeed, sexual fantasies play a major role in influencing later sexual behavior, in reflecting past experiences and these are a core variable in the systematic study of sexual identity and sexuality [7].

Italian society has generally less favourable attitudes towards unions that differ from the traditional wedding [8]. This could be due to the presence of the Catholic Church [9]. 

Furthermore, although teenagers have the tendency to have their first sexual relationship earlier than had been reported in the past, in Italy it has been observed a decrease of marriage and birthrate [10]. 

All these considerations may arise some questions about the social background of Italy and the influence on many aspects of sexuality, including internet pornography, sex toys and sexual orientation. 

Interestingly, Ross et al. [11] showed that between participants who reported using the internet to retrieve information on sexuality, younger participants displayed higher use of the medium compared to older participants, as well as bisexual men compared to heterosexual men, and males compared to females, respectively, suggesting as internet may facilitate sexual fantasies. Moreover, Daneback and Löfberg [12] suggested that using internet facilitates the expression as well as the engagement of individuals in new experiences, to a degree that would normally be not tolerated.

Finally, The COVID-19-related lockdown has profoundly changed human behaviors and habits, impairing general and psychological well-being with psychosocial consequences on sexual behavior. Jannini et al. demonstrated that anxiety and depression scores were significantly lower in subjects sexually active during lockdown [13]. In particular, sexual activity, and living without partner during lockdown as significantly affecting anxiety and depression scores [13]. 

Based on all these premises, the scope of web survey was to collect information about sexual habits and behaviours of Italian people of all ages, genders and sexual orientations and to describe patterns of sexual behaviour. 

## 2. Materials and Methods

A quantitative correlational research design was implemented for this study to evaluate the sexual habits in Italian participants in all gender and sexual interest. The study was conducted from 1 June 2019 to 31 December 2019. 

Participants were selected through posts on social networks (Instagram and Facebook) and the survey was developed and administered online through Google Forms. Each participant gave the consent to complete the study. 

Basic demographic information was collected: gender, age, height, weight, smoking habit, place of residence, sexual orientation, education level, religion and relationship status and duration.

After that, participants completed a survey with questions about their sexual habits. The questions evaluated a variety of aspects: frequency and pleasantness experienced when being involved in various sexual activities (self-stimulation, being masturbated by the partner, masturbate the partner, receiving and giving oral sex, vaginal penetration, receiving and giving anal penetration), sexual satisfaction, frequency of orgasm, stimuli used to get aroused during auto-eroticism, the use of sex toys, pleasantness of various sexual fantasies, pornography use, betrayal, traumatic sexual experiences, stress, contraception, protection against sexually transmitted infections, use of medications or drugs, use of dating apps or sites and sexting. 

The survey was conducted in Italian according to the Checklist for Reporting Results of Internet E-Surveys [14]. 

All the study procedures were carried out in accordance with the Declaration of Helsinki (2013) of the World Medical Association. The survey was anonymous and participants provided their consent to participate.

### Statistical Analysis

The qualitative data was tested using the chi-square test or Fisher’s exact test, where appropriate, while the continuous variables, presented as median (interquartile range [IQR]), were tested using Mann-Whitney U-Test or Student t test according to their distribution (according to the Kolmogorov-Smirnov test). For all statistical comparisons, significance was considered as *p* < 0.05.

## 3. Results

### 3.1. Characteristics of Participants

The median (IQR) age was 20 (18–23) years. Most of the participants enrolled were females, with 7719 (61.3%) individuals, men were 4805 (38.2%), whereas Trans were 20 (0.2%). Participants were stratified by Area of Origin, with 6036 (47.9%) coming from Northern Italy, 2646 (21.0%) from the Center and 3908 (31.0%) from the South and Islands of Italy. The education level was Higher in 7481 (59.4%) of people, with university degrees in 4211 (33.4%). Heterosexual were the most represented participants, with 10,153 (80.6%) people, Homosexual were 234 (1.9%), Bisexual 2087 (16.6%) and Pansexual 83 (0.7%). 2512 (20.0%) participants reported not having a partner, 1325 (10.5%) having occasional partners, 8598 (68.3%) having a stable relationship, 155 (1.2%) having Polyamorous relationships. The median (IQR) duration of relationships was 15 (6–36) months. 12,152 (96.5%) of our participants has no children. The median (IQR) age of the first sexual experiences was 15 (14–17) years whilst the age of the first sexual intercourse was 17 (15–18). Table 1 lists the baseline characteristics of the patients. 

### 3.2. Sexual Experience

We questioned responders about their sexual behaviors and we investigated the frequency of each experience by dividing them into “Never”, “Few times a year”, “About once a month”, “About once a week”, “Several times a month “,” Several times a week “,” Several times a day “and” Every day “. These results are shown in Table 2. 

### 3.3. Sex Toys, Sexual Pleasure and Pornography

We asked which sex toys were used during autoeroticism. We investigated what brought pleasure and arousal. The answers were expressed according to the Likert scale, where 1 indicates “not pleasure”, 2 indicates “a little pleasure”, 3 indicates “enough pleasure”, 4 indicates “very pleased” and 5 indicates “maximum pleasure”. These results are shown in Appendix A. Table 3 shows results of sex toys usage, types and use frequency. 

With our survey, we investigated the use of pornography, by asking in which context, how often and which type of pornographic material they prefer to use (Table 4). 

At the chi-square test we demonstrated that heterosexuals, homosexuals and bisexuals and were more likely to watch porn more than several times a week (40.9%, 56.4% and 47.4%) respect to Demi (24%), Queer (40.9%) or Pansexual (39.7%) (*p* < 0.01). 

The rate of watching lesbian among male, female and other were 22.5%, 77.1% and 0.4% respectively, while for 18 years old/Teenagers/Young category they were 80.9%, 18.9% and 0.2% respectively. 

### 3.4. Contraception and Frequency of Sexual Intercourse

We also investigated the use of contraception, the frequency of sexual intercourse and masturbation and the frequency of reaching orgasm (Appendix A). 

### 3.5. Use of Substances and Dating App

We investigated the possible use of exciting substances, drugs and substances that increase sexual potency (Table 5). 

Male were more likely to use drugs always than other categories (90.9% vs. 1.1%; *p* < 0.01) and similar heterosexuals (81.8% vs. 9.2%; *p* < 0.01). 

Finally, we investigated the use of social networks or dating sites to find partners with whom to have sex, and we asked what type of material was exchanged on these platforms. The results are indicated in Appendix A. 

## 4. Discussion

Our survey has investigated contemporary sexual behaviour in Italy before COVID-19 pandemic in most of its forms, taking into consideration any gender and sexual orientations. When it comes to sex, there is always plenty of curiosity, but, at the same time, often reticence and embarrassment. 

Very often, the interest in the sexual habits of the healthy general population is not properly studied. Most of the studies in the literature, in fact, focus on various pathologies of interest and on specific clinical outcomes. However, it is also important to have information on the sexual habits of the healthy general population in order to be able to establish future educational measures but also to provide data on the potential economic impact of the world regarding “sex”. Furthermore, considerable interest in the psychological aspects of sexual dysfunctions is also emerging from some international guidelines [15].

Given that the pleasure that sexual pleasure is a fundamental component of sexual health, devices designed to enhance and diversify sexual pleasure could be particularly useful in clinical practice. Despite their growing popularity and widespread use in various biopsychosocial circumstances, many taboos still seem to exist, as indicated by the paucity of scientific literature on the prevalence, application and effectiveness of sexual devices for therapeutic use [16]. 

Interestingly, compared to our European fellows, the use of sex toys in Italy is not widespread. In fact, a study of Döring et al. (2019) showed that in Germany about 50% of the respondents reported using sex toys both when masturbating and in presence of a partner [1]. Instead, our survey showed that only between 20 and 30% of the participants use sex toys and the preferred ones seem to be plugs. The fantasies that most stimulate and excite our participants are having sex in public, having sex with more than one person at the same time, blindfolded sex, being tied up and observing a naked person. 

As concerning anal sex, about 80–90% of the respondents answered that they did not practice it. A study by Habel et al., conducted in the general population of the United States between 2011–2015, reported that the prevalence of anal sex among heterosexual people is between 33–38%, which is slightly higher, compared to previous years [17]. It is possible that, in Italy, sexual freedom and the desire to experiment with new sexual practices might be overshadowed by a common feeling of shame. However, we have to consider the high rate of heterosexual respondents in our study. 

As for pornography, our results are in line with the study of Herbenick et al. [5], with 80% of our participants watching porn at home, alone or from their smartphones. 

Previously, we have demonstrated a positive association between porn addiction and erectile function, suggesting that a normal balancing between sexual activity, masturbation and pornography [18]. 

Furthermore, the impact of pornography on sexual behaviour is extremely important, expecially during previous COVID-19 pandemic. 

In fact, different studies demonstrated an increased interest in pornography and coronavirus-themed pornography after the outbreak of COVID-19 in both eastern and western countries [19,20].

All these data reflect the development of faster internet connections and the pervasive distribution of smartphones, that have somehow partially replaced the use of larger computers and devices, making pornography even more easily and discreetly accessible from everywhere, at any time, and its importance in the context of sexual behaviour. 

The most worrying data that arises from our survey is that only 66% of the respondents use condoms and only 37% use them regularly, when in a stable relationship. Since in our cohort the rate of occasional partners (10.5%) and polyamorous relationship (1.2%) were low, we believe is fundamental that new and more incisive awareness campaigns should be carried out, in order to avoid the spread of sexually transmitted diseases. 

Strenght of the current study are represented by the inclusion of a large number of participants and to have investigated different aspect of sexuality and sexual behaviour. Our results could be useful for further researches in the field and to have a photography of sexual behaviuour of a young Italian population

Limitations of our study include the lack of investigation of older people, due to the use of Internet as source of enrolment, and the use of not standarized questionnaires. Futhermore, sexual habits and behaviours of may be different after COVID-19 pandemic and they should be taken into account by future researches. Finally, our cohort was young and it may be not representative of the general population. 

## 5. Conclusions

Our survey was born with the aim of gaining a representative picture of sexuality in Italy before COVID-19 pandemic. There is still much to be done in order to increase people’s awareness of sexual pleasure and get to the point of feeling free to express sexual desires to a partner, without fearing to be judged. But even more important, it is necessary to increase awareness campaigns for the prevention of sexually transmitted diseases, especially among young people, who are more at risk, since they have fewer stable relationships and therefore often relate with different sexual partners. Moreover, our results can have several practical implications for the areas of sex education, sexual health and to counteract sexual dysfunction during COVID-19 pandemic. Given the current trends of sexual habits, it is necessary to safeguard the health of young people and support them by increasing their sexual well-being.

## Figures and Tables

**Table 1 jcm-10-04327-t001:** Basic characteristics of the participants of our study.

Participants, *n* = 12,590	
Age, years median (IQR)	20 (18–23)
Height, cm median (IQR)	170 (163–177)
Weight, kg median (IQR)	64 (55–74)
BMI, kg/m^2^ median (IQR)	22.1 (20.2–24.7)
Gender, *n* (%)	
Male	4805 (38.2)
Female	7719 (61.3)
Trans	20 (0.2)
Other	42 (0.3)
Area of Origin, *n* (%)	
Northern	6036 (47.9)
Center	2646 (21.0)
South and Islands	3908 (31.0)
Education level, *n* (%)	
Primary education	4 (0.1)
Secondary education	894 (7.1)
Higher education	7481 (59.4)
Universities	4211 (33.4)
Religion, *n* (%)	
Atheist	4908 (40.9)
Agnostic	1142 (9.5)
Believer	5931 (49.5)
Smoking, *n* (%)	
Yes	4553 (36.2)
No	8037 (63.8)
Sexual Orientation, *n* (%)	
Heterosexual	10,153 (80.6)
Homosexual	234 (1.9)
Bisexual	2087 (16.6)
Demi	25 (0.2)
Queer	7 (0.1)
Pansexual	83 (0.7)
Type of relationship, *n* (%)	
No partner	2512 (20.0)
Occasional partners	1325 (10.5)
Stable relationship	8598 (68.3)
Polyamorous relationship	155 (1.2)
Time of the relationship, months median (IQR)	15 (6–36)
Children, *n* (%)	
Yes	438 (3.5)
No	12,152 (96.5)
First sexual experiences, age median (IQR)	15 (14–17)
First sexual intercourse, age median (IQR)	17 (15–18)

**Table 2 jcm-10-04327-t002:** Sexual experience patterns in the total cohort.

Participants, *n* = 12,590	
Partner Masturbates You	
Responders, *n* (%)	
Never	774 (6.1)
Few times a year	571 (4.5)
About once a month	704 (5.6)
About once a week	2701 (21.5)
Several times a month	1745 (13.9)
Several times a week	5258 (41.8)
Several times a day	285 (2.3)
Every day	552 (4.4)
You masturbate your partner	
Responders, *n* (%)	
Never	613 (4.9)
Few times a year	429 (3.4)
About once a month	589 (4.7)
About once a week	2641 (21.0)
Several times a month	1749 (13.9)
Several times a week	5638 (44.8)
Several times a day	336 (2.7)
Every day	595 (4.7)
Partner practices oral sex on you	
Responders, *n* (%)	
Never	1298 (10.3)
Few times a year	918 (7.3)
About once a month	1056 (8.4)
About once a week	2550 (20.3)
Several times a month	2053 (16.3)
Several times a week	4182 (33.2)
Several times a day	190 (1.5)
Every day	343 (2.7)
You practice oral sex on the partner	
Responders, *n* (%)	
Never	967 (7.7)
Few times a year	640 (5.1)
About once a month	874 (6.9)
About once a week	2526 (20.1)
Several times a month	2158 (17.1)
Several times a week	4752 (37.7)
Several times a day	235 (1.9)
Every day	438 (3.5)
Vaginal penetrative intercourse	
Responders, *n* (%)	
Never	1117 (8.9)
Few times a year	446 (3.5)
About once a month	656 (5.2)
About once a week	2450 (19.5)
Several times a month	1620 (12.9)
Several times a week	5366 (42.6)
Several times a day	401 (3.2)
Every day	534 (4.2)
Anal penetrative intercourse (inseritive)	
Responders, *n* (%)	
Never	10,310 (81.9)
Few times a year	1009 (8.0)
About once a month	377 (3.0)
About once a week	230 (1.8)
Several times a month	375 (3.0)
Several times a week	243 (1.9)
Several times a day	22 (0.2)
Every day	24 (0.2)
Anal penetrative intercourse (receptive)	
Responders, *n* (%)	
Never	9896 (78.6)
Few times a year	1356 (10.8)
About once a month	471 (3.7)
About once a week	213 (1.7)
Several times a month	423 (3.4)
Several times a week	193 (1.5)
Several times a day	23 (0.2)
Every day	15 (0.1)
Autoeroticism	
Responders, *n* (%)	
Never	1344 (10.7)
Few times a year	834 (6.6)
About once a month	799 (6.3)
About once a week	1340 (10.6)
Several times a month	1361 (10.8)
Several times a week	4272 (33.9)
Several times a day	789 (6.3)
Every day	1851 (14.7)

**Table 3 jcm-10-04327-t003:** Sex toys use in the total cohort.

Participants, *n* = 12,590	
What do you use to get excited during autoeroticism	
Responders, *n* (%)	
Videos	2802 (22.2)
Sextoys	17 (0.1)
Erotic fantasies	5820 (46.2)
Erotic narrative	980 (7.8)
Erotic images	1573 (12.5)
Nothing	69 (0.5)
I don’t practice it	1277 (10.1)
How often do you use sex objects/toys during sexual intercourse	
Responders, *n* (%)	
I don’t have sex	683 (5.4)
Ever	8511 (67.6)
Few times	2502(19.9)
About half the time	462 (3.7)
Many times	352 (2.8)
Always	80 (0.6)
How often do you use sex objects/Toys during masturbation?	
Responders, *n* (%)	
I don’t have sex	372 (3.0)
Ever	9266 (73.6)
Few times	1731 (13.7)
About half the time	379 (3.0)
Many times	441 (3.5)
Always	401(3.2)
What kind of sex toys do you use most frequently?	
Responders, *n* (%)	
Vibrating rings	8 (0.1)
Fruit/vegetables	1048 (8.3)
Cock-rings	114 (0.9)
Sexy underwear	853 (6.8)
Disguise	1013 (8.0)
Fetish objects	1613 (12.8)
Lubricant	95 (0.8)
Dildos	113 (0.9)
Butt plung/anal dilators	7648 (60.8)
Objects for daily use	21 (0.2)
Strap-ons	11 (0.1)
Vibrator	10 (0.1)
Balls	1 (0.0)
Fleshlight (artificial vaginas)	15 (0.1)
I don’t use it	26 (0.1)

**Table 4 jcm-10-04327-t004:** Pornography Patterns.

Participants, *n* = 12,590	
In general, in which context do you see pornography most frequently?	
Responders, *n* (%)	
In pairs	503 (4.0)
Alone	10,128 (80.5)
In a group	11 (0.1)
Never	1942 (15.4)
How often do you view online pornography?	
Responders, *n* (%)	
Never	1816 (14.4)
Few times a year	1512 (12.0)
About once a month	1015 (8.1)
About once a week	1468 (11.7)
Several times a month	1457 (11.6)
Several times a week	3501 (27.8)
Many times a day	527 (4.2)
Everyday	1294 (10.3)
What is the most frequent topic of the pornographic material you use?	
Amateur	342 (5.5)
Anal	242 (3.9)
Asian	22 (0.4)
Masturbation	143 (2.3)
Bbw	14 (0.2)
Bdsm	215 (3.5)
Big Ass	26 (0.4)
Big Boobs	85 (1.4)
Big Cock	5 (0.1)
Blonde	19 (0.3)
Bisexual	15 (0.2)
Black	18 (0.3)
Oral Sex	167 (2.7)
Bondage	111 (1.8)
Brazzers	9 (0.1)
Casting	28 (0.5)
Lesbian	734 (11.9)
Classic	106 (1.7)
Compilation	17 (0.3)
Cunnilingus	39 (0.6)
Couple	67 (1.1)
Cowgirl	7 (0.1)
Creampie	65 (1.0)
Cuckold	16 (0.3)
Cumshot	25 (0.4)
Curvy	6 (0.1)
Deepthroat	20 (0.3)
18 years old/Teenagers/Young	572 (9.2)
It Depends	58 (0.9)
Domination	13 (0.2)
Style Stays	12 (0.2)
Double Penetration	45 (0.7)
Doctor	6 (0.1)
Threesome	423 (6.8)
Ebony	12 (0.2)
Erotic	27 (0.4)
Straight	328 (5.3)
Facesitting	14 (0.2)
Fak Taxi Familia/Incesti/Daddy/Stepsister	16 (0.3)
Fantasy	250 (4.0)
Fendom	8 (0.1)
Fetishism	8 (0.1)
Fingering	34 (0.5)
Footjob	13 (0.2)
Gang Bang	34 (0.5)
Fisting	103 (1.7)
Cartons	12 (0.2)
Gay	13 (0.2)
Role Play Games	184 (3.0)
Hardcore	6 (0.1)
Hentai	165 (2.7)
Handjob	95 (1.5)
Several Racies	5 (0.1)
Italian	19 (0.3)
Massage	21 (0.3)
Mature/Milf	66 (1.1)
None In Particular	440 (7.1)
Orgasm	102 (1.6)
Squirting	55 (0.9)
Group Sex	62 (1.0)
Passionate/Romantic	135 (2.2)
Betrayal	14 (0.2)
Transsexual	9 (0.1)
Spanking	17 (0.3)
Public Sex	50 (0.8)
Red Head	16 (0.3)
Pissing	10 (0.2)
Pov	47 (0.8)
Other	110 (1.8)
Where do you view online pornography?	
Responders, *n* (%)	
I don’t watch porn	1845 (14.7)
At home	10,721 (85.2)
At work	17 (0.1)
In public	7 (0.1)
Which device do you most frequently view pornography with?	
Responders, *n* (%)	
I don’t watch porn	1851 (14.7)
Computers	1031 (8.2)
Video game consoles	18 (0.1)
Smartphones	9294 (73.8)
Tablets	396 (3.1)

**Table 5 jcm-10-04327-t005:** Use of Substances among the general cohort.

Participants, *n* = 12,590	
How frequently do you use the following drugs in sexuality?	
Viagra, Cialis, Levitra, Spedra	
Responders, *n* (%)	
Never	12,412 (98.8)
Hardly ever	94 (0.7)
Sometimes	32 (0.3)
Often	8 (0.1)
Always	11 (0.1)
Paroxetine (Daparox/Eutimil), Priligy, Drops/Creams for premature ejaculation	
Responders, *n* (%)	
Never	12,433 (99.0)
Hardly ever	78 (0.6)
Sometimes	25 (0.2)
Often	9 (0.1)
Always	16 (0.1)
Alcohol	
Responders, *n* (%)	
Never	6905 (54.9)
Hardly ever	3207 (25.5)
Sometimes	1851 (14.7)
Often	527 (4.2)
Always	86 (0.7)
Stimulants (Cocaine, Amphetamines etc.)	
Responders, *n* (%)	
Never	12,371 (98.4)
Hardly ever	131 (1.0)
Sometimes	42 (0.3)
Often	15 (0.1)
Always	10 (0.1)
Relaxing (Cannabis etc.)	
Responders, *n* (%)	
Never	9910 (78.8)
Hardly ever	1273 (10.1)
Sometimes	764 (6.1)
Often	431 (3.4)
Always	195 (1.6)
Hallucinogens	
Responders, *n* (%)	
Never	12,464 (99.2)
Hardly ever	68 (0.5)
Sometimes	19 (0.2)
Often	8 (0.1)
Always	9 (0.1)
What reasons push you to use these substances?	
Responders, *n* (%)	
Habitual Use	38 (0.3)
Funny	52 (0.4)
Occasional Use	201 (1.6)
It Like Me	81 (0.6)
Improves Performance	449 (3.6)
Improves Sensations	1400 (11.2)
To Decrease Performance Anxiety	400 (3.2)
To Eliminate The Inhibitor Brakes	1067 (8.6)
For Transgression	537 (4.3)
Relaxation	49 (0.4)
Increase Excitement	16 (0.1)
I Don’t Use It	8098 (64.9)
Other Reasons	82 (0.7)
How do you rate sexuality using these substances?	
Responders, *n* (%)	
I Don’t Use Any Substance	7976 (63.8)
Not At All Satisfactory	127 (1.0)
Unsatisfactory	418 (3.3)
Quite Satisfactory	2420 (19.3)
Very Satisfying	1570 (12.5)

## Data Availability

Data are available upon request.

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
