# Peer review of "Sexual Behaviour and Fantasies in a Group of Young Italian Cohort"

_jcm, 2021, doi:10.3390/jcm10194327_

Round 1

Reviewer 1 Report

The paper has greatly improved, but I still have some concerns.

Given the study was conducted prior to COVID-19, it seems odd to provide information on sex and sexuality in Italian culture during the pandemic, with no information about sexual norms prior. If this study is read globally, some information on Italian culture would be useful. For example, is Italy known as a relatively liberal country regarding sex? Given the broad focus of the questions asked in this study, some general background on Italian sexual culture could be included. Tortora et al. (2019) is included, but are there any more studies? Also, Jannini et al. is not in the reference list.

Related, adding in that sexuality has profoundly changed during the pandemic makes the point about how relevant is the current study anymore? Arguably, the sexual habits and behaviours of your participants may be completely different now (according to your own statement). Further acknowledgement of this is needed.

The authors state “Our strength however, was to take a photograph of a great number of patients under examination in order to have a starting point in our day for further studies and considerations.” Rereading the method section, it doesn’t mention that participants were patients. Presumably this is a miscommunication in the response letter, or were the participants also patients of some sort?

While I agree a photograph of the sexual behaviours of participants is useful, there could still be some further analyses for this paper. There are ways of performing analyses without a binary variable, e.g., a Chi-Square could be used to explore pornography use of the different genders.

The discussion is much improved.

Author Response

Reviewer's 1 comments:

The paper has greatly improved, but I still have some concerns.

  1. Given the study was conducted prior to COVID-19, it seems odd to provide information on sex and sexuality in Italian culture during the pandemic, with no information about sexual norms prior. If this study is read globally, some information on Italian culture would be useful. For example, is Italy known as a relatively liberal country regarding sex? Given the broad focus of the questions asked in this study, some general background on Italian sexual culture could be included. Tortora et al. (2019) is included, but are there any more studies? Also, Jannini et al. is not in the reference list.
  2. Related, adding in that sexuality has profoundly changed during the pandemic makes the point about how relevant is the current study anymore? Arguably, the sexual habits and behaviours of your participants may be completely different now (according to your own statement). Further acknowledgement of this is needed.
  3. The authors state “Our strength however, was to take a photograph of a great number of patients under examination in order to have a starting point in our day for further studies and considerations.” Rereading the method section, it doesn’t mention that participants were patients. Presumably this is a miscommunication in the response letter, or were the participants also patients of some sort?
  4. While I agree a photograph of the sexual behaviours of participants is useful, there could still be some further analyses for this paper. There are ways of performing analyses without a binary variable, e.g., a Chi-Square could be used to explore pornography use of the different genders
  5. The discussion is much improved.

Answer to reviewer

We would like to thank the reviewer for his time and his suggestions to improve the manuscript.

  1. Thank you for this important comment. We have added in the introduction a background on Italian culture. The reference of Jannini has been added. We have also modified the introduction adding more information about Italian culture.
  2. Yes, you are totally right. We put a sentence in the limitation.
  3. We apologize for this misunderstanding. We did not collect data from patients but only from participants. We modified the typo in the text.
  4. We added a further analysis investigating the impact of pornography on sexual orientation. In particular, At the chi-square test we demonstrated that heterosexuals, homosexuals and bisexuals and were more likely to watch porn more than several times a week (40.9%, 56.4% and 47.4%) respect to Demi (24%), Queer (40.9%) or Pansexual (39.7%) (p<0.01). We added these results in the text.
  5. Thank you so much for your comments.

Reviewer 2 Report

I suggest some linguistic and content-related improvements, e.g.:

Introduction

  • line 34 “… in what in”
  • “ Sexual behavior has changed dramatically over the years” -> In what sense dramatically? Do you mean that sexual intercourse takes place later than in the past, higher pornography consumption, …? Please clarify.
  • line 48: A recent survey by 48 Herbenick et al. (2020), -> no comma
  • Please clarify why there is a particular interest in Italien people: are there profound differences in sexual behavior in relation to other nations? The origin of the authors is not sufficient to justify the sample - please add some more information.

Materials & Methods

  • it would be interesting to know sexual orientation in relation to gender (e.g., how many trans* were hetero/homo/bi?)
  • the same on relationship status, gender & sexual orientation -> Maybe the table could be restructured accordingly?

Results

  • I recommend writing down the main findings under the tables in 1-2 short sentences.
  • For some results, additional information on age ranges, sexual orientation, and relationship status would be interesting (e.g., drug use during sex. activities, use of lesbian porn & young/underaged porn -> predominantly watched by (heterosexual?) males or also by females?)

Discussion

Please re-check the paper for careless mistakes and English grammar, there are a few mistakes regarding spelling and grammar.

  • line 171: It is interesting to address that given that sexual pleasure is a core component of sexual health, devices that are designed to enhance and diversify sexual pleasure are particularly useful in clinical practice
  • Regarding sex toys and the comparison with Germany, it should be noted that this Italian sample is very young and therefore perhaps not representative for making international comparisons. According to current studies, fewer sex toys are generally used in this age group. The same with the topic of anal sex.
  • line 189: However, we have to consider the high rate of eterosexual respondents in our study
  • line 196: Furthermore, the impact of pornography on sexual behaviour is estremely important, expecially during previous COVID-19 pandemic.
  • line 211: and the not standardized application of questionnaires -> check grammar
  • I don’t understand the conclusion of the discussion: “In conclusion, our survey has shown that, although nowadays it is more convenient and easier to meet people through social networks and dating sites, the best way to meet partners seem to remain the old fashioned "through mutual friends” -> The reference to the object of investigation and the research question is missing. Is this rather a personal opinion of the authors?
  • Much more interesting would be a comparison with sexual behavior DURING the covid pandemic: are there studies showing differences? Were there more sex or other pornography issues before the pandemic? For example, it is now known that new porn genres such as "covid-porn" have come onto the market as a result of the pandemic. You collected data on sexuality just before the pandemic, surely this could be well used to make a comparison of the sexual behavior of Italians pre-post-pandemic.

To put it in a nutshell, really interesting data was collected. The discussion should be revised to emphasize the relevance of the study. There are a few more suggestions that we recommend to implement.

Author Response

Reviewer's 2 comments:

Suggest some linguistic and content-related improvements, e.g.:

Introduction

  1. line 34 “… in what in”
  2. “ Sexual behavior has changed dramatically over the years” -> In what sense dramatically? Do you mean that sexual intercourse takes place later than in the past, higher pornography consumption, …? Please clarify.
  3. line 48: A recent survey by 48 Herbenick et al. (2020), -> no comma
  4. Please clarify why there is a particular interest in Italian people: are there profound differences in sexual behavior in relation to other nations? The origin of the authors is not sufficient to justify the sample – please add some more information.
  5. it would be interesting to know sexual orientation in relation to gender (e.g., how many trans* were hetero/homo/bi?)
  6. the same on relationship status, gender & sexual orientation -> Maybe the table could be restructured accordingly?
  7. I recommend writing down the main findings under the tables in 1-2 short sentences.
  8. For some results, additional information on age ranges, sexual orientation, and relationship status would be interesting (e.g., drug use during sex. activities, use of lesbian porn & young/underaged porn -> predominantly watched by (heterosexual?) males or also by females?)
  9. Please re-check the paper for careless mistakes and English grammar, there are a few mistakes regarding spelling and grammar.
  10. line 171: It is interesting to address that given that sexual pleasure is a core component of sexual health, devices that are designed to enhance and diversify sexual pleasure are particularly useful in clinical practice. Regarding sex toys and the comparison with Germany, it should be noted that this Italian sample is very young and therefore perhaps not representative for making international comparisons. According to current studies, fewer sex toys are generally used in this age group. The same with the topic of anal sex.
  11. line 189: However, we have to consider the high rate of eterosexual respondents in our study
  12. line 196: Furthermore, the impact of pornography on sexual behaviour is estremely important, expecially during previous COVID-19 pandemic.
  13. line 211: and the not standardized application of questionnaires -> check grammar
  14. I don’t understand the conclusion of the discussion: “In conclusion, our survey has shown that, although nowadays it is more convenient and easier to meet people through social networks and dating sites, the best way to meet partners seem to remain the old fashioned "through mutual friends” -> The reference to the object of investigation and the research question is missing. Is this rather a personal opinion of the authors?
  15. Much more interesting would be a comparison with sexual behavior DURING the covid pandemic: are there studies showing differences? Were there more sex or other pornography issues before the pandemic? For example, it is now known that new porn genres such as "covid-porn" have come onto the market as a result of the pandemic. You collected data on sexuality just before the pandemic, surely this could be well used to make a comparison of the sexual behavior of Italians pre-post-pandemic.
  16. To put it in a nutshell, really interesting data was collected. The discussion should be revised to emphasize the relevance of the study. There are a few more suggestions that we recommend to implement.

Answer to reviewer

We would like to thank the reviewer for his time and his suggestions to improve the manuscript.

  1. We corrected the typo.
  2. We agree with our comment and we changed the word into “consistently” that is more appropriate.
  3. We corrected the typo
  4. We have added some sentences about Italian culture that may justify our background. In particular, Italian society has generally less favourable attitudes towards unions that differ from the traditional wedding. This could be due to the presence of the Catholic Church. Furthermore, although teenagers have the tendency to have their first sexual relationship earlier than had been reported in the past, in Italy it has been observed a decrease of marriage and birthrate. All these considerations may arise some questions about the social background of Italy and the influence on many aspects of sexuality, including internet pornography, sex toys and sexual orientation.
  5. We would like to thank you for this comment. We have data on this, however the association between gender (male, female, trans or other) and sexual orientation could be misunderstood and since that it is better to deal about sexual fluidity, a potential significant p-value could not add more insights.
  6. We think of the same as above. Eventually we have the table for it, but saying that trans were more likely to be bisexual could be not appropriate for the concept of sexual fluidity.
  7. We would like to thank you for this comment. As for Table 2, it reports many percentages and we would like to maintain a certain grade of simplification since the paper is already very full of data. There are no specific data to report in the sentences since all could be important for the reader.
  8. We have added more results in the paper.
  9. We corrected all typos.
  10. We have also added the limitation regarding the young age of our cohort.
  11. We modified the conclusion.
  12. We modified the discussion according to your suggestions.